# Effect of Slightly Acidic Electrolyzed Water on Growth, Diarrhea and Intestinal Bacteria of Newly Weaned Piglets

**DOI:** 10.3390/genes14071398

**Published:** 2023-07-04

**Authors:** Xiaoxia Hao, Dan Xie, Dongmei Jiang, Li Zhu, Linyuan Shen, Mailin Gan, Lin Bai

**Affiliations:** 1College of Animal Science and Technology, Sichuan Agricultural University, 211 Huimin Ave, Chengdu 611130, China; 14124@sicau.edu.cn (X.H.); 2020302149@stu.sicau.edu.cn (D.X.); jiangdm@sicau.edu.cn (D.J.); zhuli@sicau.edu.cn (L.Z.); shenlinyuan@sicau.edu.cn (L.S.); ganmailin@stu.sicau.edu.cn (M.G.); 2Farm Animal Genetic Resource Exploration and Innovation Key Laboratory of Sichuan Province, Sichuan Agricultural University, Chengdu 611130, China

**Keywords:** drinking water, slightly acidic electrolyzed water (SAEW), weaned piglets, intestinal microbiomes, diarrhea rate

## Abstract

As an environmentally-friendly agent, slightly acidic electrolyzed water (SAEW) was introduced in drinking water of newly weaned piglets for diarrhea prevention. In total, 72 piglets were employed and 3% SAEW was added into the normal temperature and warm (30 °C) tap water, respectively, for this 33-day feeding experiment. It was found that the total bacteria and coliforms in the drinking water were reduced by 70% and 100%, respectively, with the addition of 3% SAEW. After SAEW treatment, the average daily water and feed intakes of piglets were increased during the first 16 days, and the diarrhea rate was reduced by 100%, with not one case of diarrhea recorded at the end of the experiment. The microbiome results demonstrated that SAEW decreased the diversity of caecum bacteria with normal tap water supplied, and increased the richness of the caecum bacteria with warm tap water supplied. SAEW also increased the abundance of potentially beneficial genera *Sutterella* and *Ruminococcaceae_UCG-005* and reduced the abundance of pathogenic *Faecalibacterium*. Moreover, twelve metabolic functions belonging to the cluster of metabolism and organismal functions, including digestion and the endocrine and excretory systems, were greatly enhanced. Correlation analysis indicated that the influence of intestinal pathogens on water and feed intakes and the diarrhea of piglets were decreased by SAEW. The results suggest that SAEW can be used as an antibiotic substitute to prevent diarrhea in newly weaned piglets.

## 1. Introduction

The development of alternatives to antibiotics has been an important issue in the swine industry as using antibiotics as growth promoters in livestock feeds has been banned in the EU since 2006 and in China since 2020. Since then, administering prebiotics and probiotics as water additives in order to modulate the intestinal microbiota of livestock is considered an effective health strategy [1]. Furthermore, the provision of sufficient amounts of quality drinking water plays an important role in maintaining health and performance in food-producing animals [2]. Therefore, drinking water quality should be guaranteed in order to ensure optimal animal health [3]. The total *E. coli* populations, less than 100 MPN/100 mL and 10 MPN/100 mL in adult and young livestock drinking water, respectively, have been provided according to “Drinking Water Quality for Livestock and Poultry—Pollution free Food (NY5027-2008)” in China.

Slightly acidic electrolyzed water (SAEW) has gained increasing attention as a new non-thermal method for microbial inactivation [4]. The SAEW with a pH value of 5.0–6.5, contains mixed oxidants, including hypochlorous acid (HOCl), H^+^, and O^3^, and other active ingredients, such as O_2_ and Cl_2_, and it has been shown to have strong anti-microbial activity [5]. In addition, SAEW, which is produced from diluted hydrochloric acid, has minimal negative effects on human health and safety, reduces corrosion, and limits phototoxic effects while maximizing the application of hypochlorous acid on antibacterial species [6]. In this way, SAEW is 150 times more effective than sodium hypochlorite solution in disinfecting water sources [5]. This relatively new concept has been used in food, agriculture, livestock production, medical sterilization, and other areas that require the use of anti-microbial technologies [7,8]. In swine production, SAEW has proved more effective in disinfecting the air and surfaces in the barn than other chemicals [9]. At the end of the production chain, electrolyzed water showed strong bactericidal effects on pigskin microorganisms, while not affecting skin graft histological properties [10]. It also showed no cytotoxicity. However, as a drinking water additive, electrolyzed water has been understudied. Inagaki et al. [11] assessed the safety of SAEW in the drinking water of laboratory mice over 12 weeks, and no significant difference was observed in serum aspartate aminotransferase (AST), alanine aminotransferase (ALT), and creatinine levels, and they also reported no significant lesions on any collected liver, kidney, heart, lung, spleen, stomach, or esophagus samples. On the other hand, Watson et al. [12] found that with the provision of electrolyzed water, the duration of symptoms of infected pigs was markedly lessened and the severity of symptoms was reduced, and they also showed that using electrolyzed water as either a substitute for or addition to regular drinking water for pigs infected with the porcine epidemic diarrhea virus resulted in a much higher survival rate. Bodas et al. [13] showed that adding 3% SAEW to tap water significantly reduced bacterial counts (*p* < 0.05), while the blood gases of dairy ewes were not affected. Studies also showed that SAEW is effective in disinfecting the drinking water system of layer hen houses while contributing to a reduced mortality rate [14]. Even though the disinfectant power of SAEW on the drinking water of animals has been proved, its effect on the gut environment is still not clear.

The intestinal microbiota, including its beneficial, harmful, and neutral flora, plays an essential role in the maintenance of intestinal homeostasis and animal health [15]. The modification of the gut microbiota due to diets and medication, such as antibiotics, has been implicated in diseases in animal production. Post-weaning is considered one of the most delicate periods in swine production [16], as newly weaned piglets need to acclimate to the change in diet, which may greatly affect their intestinal microbiome and its associated metabolism functions. To this end, high-throughput sequencing has proven useful for assessing the state of the intestinal microbiota and metabolism of newly weaned piglets [17]. The intestinal microbiota composition and metabolism in early weaned piglets were assessed through 16S rRNA sequencing [18].

For this study, newly weaned piglets, which are sensitive to drinking water changes, were used to investigate the effect of SAEW in regular drinking water (tap water) on the piglet gut microbial population. Piglet drinking water and feed intakes as well as growth and health performance were evaluated in order to estimate the effect of SAEW on the microbial diversity and community structures in the rectum and the caecum of piglets. Piglet metabolism functions were also explored in order to probe the feasibility of SAEW in improving the intestinal microbial environment. Lastly, the association between intestinal pathogens and diarrhea in piglets was analyzed in order to evaluate the ability of SAEW to control diarrhea through gut microbiome amendment.

## 2. Materials and Methods

### 2.1. Slightly Acidic Electrolyzed Water (SAEW) Preparation

The slightly acidic electrolyzed water was produced by a SAEW generator (HOCL 0.2t, Weijie Trading Co., Ltd., Suzhou, China) with a 9% HCl solution as electrolyte. The SAEW, with a pH of 5.98, an oxidation-reduction potential (ORP) of 976 mV, and an available chlorine concentration (ACC) of 30 mg/L, was produced. The physicochemical properties of SAEW were measured before use, where the pH and ORP values were measured using a dual scale pH/ORP meter (CON60, Trans-Wiggens, Singapore) with a pH electrode (PE02; range 0.00–14.00) and an ORP electrode (ORP06; range −999–+999 mV). The ACC was determined using a digital chlorine test system (RC-2Z, Kasahara Chemical Instruments Corp., Saitama, Japan) with a detection range of 0–320 mg/L.

### 2.2. Disinfection of Drinking Water

Animal experiments were performed on an experimental pig farm located in Ya’an City (China). Drinking water samples were collected in the farm, and SAEW (30 mg/m^3^) was used to determine the disinfection efficiency of the drinking water. Preliminary experiments determined the necessary SAEW dose as 3%, and water was left to stand for 5 min after mixing. Sodium thiosulfate solution (Na_2_S_2_O_3_, 0.1%) was then added to stop the residual activity of the SAEW. Aliquots of 0.1 mL of the undiluted sample and 10-fold serial dilutions were inoculated in triplicate onto two different culture mediums, Plate count agar (PCA) for the total bacteria, and MacConkey agar (MCA) for Coliforms, and then incubated at 37 °C for 24–48 h. The colony-forming units (CFUs) on each plate were counted and three independent replicates were performed.

### 2.3. Animal Breeding and Experimental Design

A total of 72 Duroc × Landrace × Yorkshire crossbred piglets with similar weights, good health status, and no history of disease were selected for this study. The piglets were weaned at day 28, and feed with no added antibiotics was provided. Four separate units with their own water containers and drinkers were used. The SAEW was generated, mixed with tap water, and added to each container every day, which supplied the water through automatic drinking nozzles. Each nozzle was fitted with a water flow meter to record water intake. During this experiment, the routine immunization procedure was performed on all piglets (swine fever vaccine was given by intramuscular injection with a dose of 1 mL at 50 days of age). Researchers and workers applied strict hygiene practices to prevent cross-contamination. The average air and drinking water temperature inside the swine house were maintained at 14.5 °C and 8.6 °C, respectively, with a heat control system.

All the piglets were randomly assigned into four groups (18 piglets per group) and raised in separate pigpens. There were three pens (as a unit) in each group, with six piglets per pen. Different kinds of drinking water were provided for each group: tap water (TW), warm tap water (WTW, 30 °C), tap water with SAEW (TWS, 3% SAEW), and warm tap water with SAEW (WTWS, 3% SAEW). The average daily water intake (ADWI), average daily feed intake (ADFI), and average daily weight gain (ADG) were recorded for each pen throughout the feeding trial from 28 to 60 days of age. The feed consumption rate (FCR) was calculated by ADFI/ADG. During the experimental period, piglets were checked every day for any symptoms of diarrhea, and the diarrhea rate (DR, %) was calculated by the number of diarrhea cases/(total number of pigs × trial days). All experimental procedures were performed in accordance with the Institutional Review Board (IRB14124) and the Institutional Animal Care and Use Committee of the Sichuan Agricultural University under permit number DKY-B20156108.

### 2.4. Assessment of Intestinal Microbes

Piglets were humanely sacrificed after the animal experiment. The intestinal content (about 2 g) of the caecum and rectum was dissected and stored at −80 °C until processing. A total of 24 chyme samples (12 rectums, 12 cecum, and 3 samples from each group) were collected and processed for sequencing. The total DNA was extracted from the samples using the QIAamp^®^ Fast DNA Stool Mini Kit (TaKaRa, Otsu, Japan) according to the manufacturer’s instructions. The DNA was quantified using a Qubit 3.0 Fluorometer (Life Technologies, Shanghai, China). The bacterial 16S rRNA gene was amplified using primer 515F (5′-GTG CCA GCM GCC GCG GTA A-3′) and 806R (5′-GGA CTA CHV GGG TWT CTA AT-3′) in a final reaction volume of 50 mL. The PCR program was as follows: 94 °C for 3 min, followed by 30 cycles of 94 °C for 60 s, 55 °C for 60 s, and 72 °C for 90 s. A final step of 72 °C for 10 min was included. Each sample was amplified in triplicate. The PCR products were purified and then prepared for library construction and sequencing, which was performed on an Illumina HiSeq 2 × 250 platform (Novogene Bio-Information Technology Co., Ltd., Beijing, China). The 16S rRNA gene sequences generated in this study were deposited in the NCBI Sequence Read Archive (SRA) database with the accession number SRP149034.

### 2.5. Statistical Analysis

The antimicrobial potential of SAEW on drinking water was analyzed using a one-factor variance analysis. Growth performance (ADWI, ADFI, AFG, and FCR) and diarrhea rate (DR) differences between groups were tested using a one-factor ANOVA, and a two-factor analysis of variance at the 5% probability level was performed to determine the interactive influence between time and SAEW. Data are expressed as the mean (M) ± standard deviation (SD), and differences were considered significant if *p* < 0.05. All analyses were conducted in SPSS, version 20.0.

Raw sequences were demultiplexed and denoised using Usearch v9.0. Clean, per-sample FASTA files were analyzed using QIIME v1.9.0. Reads were assigned to operational taxonomic units (OTUs) at 97% thresholds. α diversity metrics, including the Shannon and Chao1 index, were computed, and their differences were analyzed by using the Mann–Whitney *U* test. To correct for differences in sequencing depth, we randomly subsampled the OTU table to a depth of 2500 sequences per sample 10 times before computing the α diversity metrics. A two-sided Welch’s *t*-test was used to identify species that were significantly represented between different groups. The relationships between water and feed intakes, diarrhea rate, and intestinal pathogenic bacteria were assessed by Pearson connection analysis and displayed in a heatmap.

## 3. Results

### 3.1. Disinfection Effect of the SAEW on Drinking Water

The parameters of the drinking water from the four groups are shown in Table 1. The pH values were around 6.8 with little difference between the groups. Significantly more (*p* < 0.05) CFUs of total bacteria and coliforms were found in the warm drinking water group (WTW) compared to the tap water group (TW). After SAEW addition, the total amount of bacteria and coliforms were reduced significantly (*p* < 0.05) in both WTWS and TWS treatments, with at least a 70% reduction in the total number of bacteria and a 100% reduction in the total number of coliforms. The bacterial count in the TWS group (25 CFU/mL) was lower than in the WTWS (90 CFU/mL) group. These results suggest that warm water (30 °C) carried more bacteria and that SAEW can be used as an efficient disinfectant.

### 3.2. Improvement of SAEW on Water and Feed Consumption, Growth and Health Performance of Piglets

The average daily gain (ADG) of piglets was increased from 0.44 kg/d/pig to 0.49 kg/d/pig after 33 days of breeding, and the feed consumption rate (FCR) was reduced from 1.74 to 1.68 when 3% SAEW was added to the drinking water, although the differences were not significant (*p* > 0.05). Piglets for the WTWS treatment group showed lower ADWI and ADFI, although differences were not significant (*p* > 0.05) (Table 2). However, significantly (*p* = 0.02) higher average daily water intake (ADWI) was observed in the TWS treatment than in the TW treatment during days 1–16 of the experiment (Figure 1a). Slightly lower ADWI was detected in the WTWS group compared to the WTW group during days 4–15 (Figure 1b). Moreover, significant differences (*p* < 0.05) in the average daily feed intake (ADFI) were observed between the TW and TWS, WTW, and WTWS groups during days 1–16 (Figure 1c,d). These results indicate that the ADWI and ADFI of piglets were increased with SAEW addition during the first 16 days of the experiment. No significant differences were seen between the groups during days 17–33 of the experiment (Appendix A).

The diarrhea rates (DR) of piglets ranged between 1.35% in the TW group to 0.51% in the WTW group (Table 2). Diarrhea cases were reduced by 100% with SAEW addition in both the TWS and WTWS treatments, where not one case of diarrhea was recorded. Data suggest that both water temperature and SAEW addition contributed to the DR reduction.

### 3.3. Effect of SAEW on Intestinal Bacterial Diversity

A total of 1,606,041 reads were produced. After chimera checking and filtering out singleton OTUs, a total of 1,584,614 clean reads were retained in the dataset. On average, each sample had 66,918 reads.

The α diversity showed the richness (Chao1) and diversity (Shannon) of bacteria in the rectum and caecum of piglets from the four groups. The indexes revealed significantly (*p* < 0.05) more diverse caecum microbiomes in the WTWS treatment compared with the WTW treatment (Figure 2b), while the richness of the caecum microbiome in the TWS group was significantly (*p* < 0.01) lower than that in the TW group (Figure 2d). With the addition of SAEW, a higher diversity and richness of bacteria was observed in the caecum of the WTW group compared to the TW group (*p* < 0.05, Figure 2b; *p* < 0.01, Figure 2d). This suggests that SAEW addition decreased the diversity of caecum microbiomes when tap water was supplied, and increased the richness of the caecum microbiome when warm tap water was supplied. However, no obvious α diversity differences in bacteria were observed in the feces (rectum samples) (Figure 2a,c).

### 3.4. Effect of SAEW on Intestinal Pathogenic Bacteria

A total of 36 bacterial phyla were identified in all samples, of which the Bacteroidetes, Firmicutes, and Tenericutes were the three dominant phyla, representing 86–94% of the total sequences (Figure 3a). Four phyla identified from the caecum samples (Saccharibacteria, Chlamydiae, Actinobacteria, and Tenericutes) differed significantly between groups due to SAEW administration. The Saccharibacteria and Chlamydiae showed less than 0.01% relative abundance, while the relative abundance of the Actinobacteria was significantly reduced (*p* = 0.037) by SAEW addition from 0.4% in the TWc group to 0.12% in the TWSc group. On the other hand, the Tenericutes increased significantly (*p* = 0.021) from 1.37% in the WTWc group to 4.43% in the WTWSc group due to SAEW addition.

The top 20 genera identified in this study are shown in Figure 3b. Differences in bacterial abundance between groups were significant (*p* < 0.05) for 27 genera in both rectum and caecum samples. The relative abundance of *Faecalibacterium* was significantly reduced (*p* < 0.01) from 0.73% in the TWc group to 0.34% in the TWSc group, while the *Sutterella* and *Ruminococcaceae_UCG-005* increased (*p* < 0.05) in the WTWSc group (0.91% and 1.36%) compared to the WTWc group (0.36% and 0.52%). We identified five genera associated with diarrhea in piglets (Figure 3c). *Clostridium_sensu_stricto_1* and *Clostridium_sensu_stricto_6* decreased in the TWSr and TWSc groups but increased in the WTWSr and WTWSc groups. An uncultured bacterium belonging to the Clostridiales_vadinBB60_group decreased in the TWSr, WTWSr, and WTWSc groups but increased in the TWSc. *Campylobacter* decreased in the WTWSr and TWSc groups but increased in the TWSr and WTWSc. *Escherichia-Shigella* decreased in the WTWSr but increased in the TWSc and WTWSc, and no obvious decrease or increase were observed in TWSr.

### 3.5. Effect of SAEW on Intestinal Bacterial Metabolic Functions

Similar predicted metabolic pathways were identified from all treatments, including metabolism (77.0–78.1%), genetic information processing (8.9–9.0%), environmental information processing (4.1–5.1%), cellular processes (3.7–4.0%%), human diseases (3.1–3.2%), and organismal systems (1.8–1.9%) (Figure 4a). The potential metabolic functions associated with intestinal health are illustrated in Figure 4b. As expected, the levels of twelve functions belonging to the cluster of metabolisms including biosynthesis of other secondary metabolites, global and overview maps, amino acids, carbohydrates, lipids, cofactors and vitamins, other amino acids, terpenoids and polyketides, nucleotide, and energy metabolisms, xenobiotics biodegradation and metabolism, and glycan biosynthesis and metabolism were significantly enhanced (*p* < 0.01) by the addition of SAEW in most treatments. Similarly, pathways associated with the digestive, endocrine, and excretory systems were also enhanced (*p* < 0.01). This indicates that the bacterial metabolic functions related to piglet health were enhanced by SAEW, which could aid in diarrhea prevention.

### 3.6. Relationship between Diarrhea Rate and Pathogenic Bacteria

A significantly negative correlation (r < −0.8, *p* < 0.05) was obtained between ADWI and the caecum bacterium *Escherichia-Shigella* and an uncultured rectum bacterium belonging to the family Clostridiales_vadinBB60_group in the TW treatment (Figure 5). A similar negative correlation (r < −0.8, *p* < 0.05) between ADWI and feces bacteria *Escherichia-Shigella* and *Campylobacter* was observed in the WTW group. This suggests that the genera *Escherichia-Shigella*, *Campylobacter,* and the uncultured bacterium (Clostridiales_vadinBB60_group) were mainly responsible for the differences in water intake between piglets from different groups. The reduced correlation in the TWS (r > −0.5) and WTWS (r > −0.6) treatments demonstrated that SAEW might regulate the water intake by affecting the abundance of these three bacteria (Figure 3).

Moreover, a negative correlation (r < −0.8, *p* < 0.05) between ADFI and caecum bacterium *Clostridium_sensu_stricto_6*, and rectum bacterium *Clostridium_sensu_stricto_1* was observed in the TW group, while similar negative correlation (r < −0.8, *p* < 0.05) between ADFI and rectum bacteria *Clostridium_sensu_stricto_1* and *Campylobacter*, and caecum bacterium *Escherichia-Shigella* was found in the WTW group. The changed correlation (r = 0.27) between ADFI and rectum bacterium *Clostridium_sensu_stricto_1* in the TWS and positive correlation (r > 0.5) between ADFI and rectum bacteria *Clostridium_sensu_stricto_1* and caecum bacterium *Escherichia-Shigella* in the WTWS indicated that the SAEW affected feed intake in piglets by adjusting the intestinal abundance of *Clostridium_sensu_stricto_1*, *Campylobacter,* and *Escherichia-Shigella*. The results are shown in Figure 1 and Figure 3.

A positive correlation (r > 0.8, *p* < 0.05) between diarrhea rate and feces *Escherichia-Shigella* and *Clostridium_sensu_stricto_1*, caecum *Escherichia-Shigella* and uncultured bacterium (Clostridiales_vadinBB60_group) were seen in groups where tap water was provided. When warm tap water was supplied, the bacteria with the strongest influence on diarrhea were *Clostridium_sensu_stricto_1* in the rectum and *Escherichia-Shigella* and *Campylobacter* in the caecum. No correlation analyses between diarrhea rate and bacteria could be performed in the SAEW treatments as no cases of diarrhea were observed here (Table 2). Collectively, the intestinal genera with the strongest influence on diarrhea in newly weaned piglets were *Escherichia-Shigella*, *Clostridium_sensu_stricto_1*, and *Campylobacter*. The effect of intestinal pathogens on diarrhea in newly weaned piglets was reduced by SAEW addition.

## 4. Discussion

The drinking water disinfection test indicated that warm water (30 °C) carried more bacteria, and 3% SAEW with an available chlorine concentration (ACC) of 30 mg/L can be used as an efficient anti-bacteria agent for the drinking water of piglets. Similar results were reported by Bodas et al. [13] and Wang et al. [14], who found the total viable cell count and the number of *E. coli* in drinking water were reduced with the addition of slightly acidic electrolyzed oxidizing water. The higher numbers of bacteria seen in warm water are attributed to the water temperature being close to the optimum growth temperature (37 °C) of bacteria. The result provided a data basis for the following research.

The improvement in the ADWI and ADFI of piglets during the first 16 days was attributed to the SAEW-treated water, which is reported to have smaller molecules, stronger activity, better solubility, more oxygen, and is easily absorbed via electrolysis. The active hydrogen could also act as a superoxide dismutase, eliminating free radicals in the body of livestock and poultry [19]. However, Hsu [20] found that some of the residual chlorine of SAEW leaves the solution as Cl_2_ gas as the water temperature increases, and the bad smell may lead to the lower ADWI and ADFI observed in the WTWS treatment piglets. No significant differences were seen between groups during the 17–33 day period (Appendix A), as the piglets were able to adapt to the different drinking waters after the first 16 days. Similar results were obtained by Silva et al. [21], who found that the addition of flavorings to the drinking water increases the piglet’s voluntary water and feed intake during the first-week post-weaning. This all aids in preparing the animal for the negative effects of the weaning process.

In this study, DR was decreased by 0.84% in warm drinking water groups, which may be attributed to an improved gut temperature. Studies have reported that heating feed with warm water in the first week after weaning increased both the growth performance as well as feed consumption of piglets [22]. Warm water was also used as a treatment for intrauterine growth-restricted piglets, where the warm water group resulted in the same body weight gain as the colostrum treatment group [23]. Ye et al. [24] also showed that SAEW can enhance the resistance of animals to infections. This suggests that SAEW could be used as an antibiotic substitute in animal production. Similar results were seen in a tilapia culture experiment, where SAEW added more improvement in inflammation and structural damage and had a stronger bacteriostatic effect than the antibiotics gentamicin and norfloxacin [24].

Moreover, SAEW decreased the diversity of caecum microbiomes when only tap water was provided, and it increased the richness of the caecum microbiome when warm tap water was provided. Similarly, a significantly lower microbiome diversity was observed by Higashimura et al. [25], who gave alkaline electrolyzed water to mice. In this study, the reduced intestinal microbiome was attributed to the disinfection efficiency of SAEW, which affects all bacteria [5,9,26]. It was particularly interesting that SAEW resulted in an increased microbiome diversity in the warm tap water treatment, which could be explained by the synergistic effect of temperature and electronic activity of the electrolyzed water.

We found that the Bacteroidetes, Firmicutes, and Tenericutes were the dominant bacteria in the intestines of piglets [17,27,28]. Moreover, a significantly lower relative abundance of Actinobacteria was seen in the cecum of SAEW-administered piglets than compared to the tap water control, which agreed with the results of Higashimura et al. [24]. Actinobacteria are considered an indicator of gut microbiome health, whereas *Bifidobacterium*, a member of the Actinobacteria, is known to be a natural growth promoter in newly weaned piglets [29]. The higher relative abundance of Tenericutes detected in the cecum of SAEW-administered piglets in the warm tap water treatment group was expected. It was reported the relative abundance of Tenericutes in the ileal digesta decreased as the dietary protein content decreased, where a moderate reduction in protein intake can benefit the gut health of pigs [30].

The SAEW had variable effects on intestinal pathogenic bacteria composition. For example, the *Sutterella* and *Ruminococcaceae_UCG-005* increased while the *Faecalibacterium* decreased. *Ruminiclostridium cellulolyticum* is known for modulation of the cellulolysis machinery in response to cellulosic materials [31] while the *Sutterella* spp. may play an important role in host metabolic adaptation to LM-A administration [32], and *Faecalibacterium* is associated with inflammatory bowel disease [33]. Reed [34] reported that *Escherichia* and *Clostridial* organisms are important pathogenic bacteria that cause diarrhea in piglets, whereas *Clostridium difficile* can release toxins that damage the inner lining of the intestinal tract. *Salmonella* spp., *Campylobacter* spp., and *E. coli* are also pathogens known to cause diarrhea in piglets [35,36]. In this study, *Clostridium_sensu_stricto_1* and *Clostridium_sensu_stricto_6* decreased in the TWSr and TWSc groups but increased in the WTWSr and WTWSc groups, while the uncultured bacterium belonging to the Clostridiales_vadinBB60_group, *Campylobacter* and *Escherichia-Shigella* only decreased in TWSc, WTWSr and WTWSr, respectively, with the addition of SAEW. The effect of SAEW on five pathogens related to diarrhea seems complicated. A case-control study revealed that known pathogens in piglets do not always cause diarrhea, but that the association between a variety of pathogens is rather to blame [36].

The relationship analysis between water and feed intakes, diarrhea rate, and intestinal pathogenic bacteria demonstrated that SAEW regulated water intake by affecting the abundance of the bacteria *Escherichia-Shigella*, *Campylobacter,* and uncultured bacterium belonging to the Clostridiales_vadinBB60_group. It increased feed intake by reducing the intestinal *Clostridium_sensu_stricto_1*, *Campylobacter,* and *Escherichia-Shigella*, and relieved diarrhea in newly weaned piglets by controlling *Escherichia-Shigella*, *Clostridium_sensu_stricto_1*, and *Campylobacter*. Furthermore, the metabolic functions belonging to the digestive, endocrine, and excretory systems, which are related to diarrhea in piglets, were enhanced significantly by the addition of SAEW. Similar results were reported by Li et al. [37], who carried out a long-term antibiotic-free breeding study on fattening pigs for animal welfare, health, and performance.

## 5. Conclusions

The results of this study demonstrated that slightly acidic electrolyzed water has the potential, as a drinking water additive, to decrease the total bacteria and coliforms in water, thereby improving the water and feed consumption and reducing diarrhea in newly weaned piglets. In this study, SAEW increased the abundance of the potentially beneficial genera *Sutterella* and *Ruminococcaceae_UCG-005* and reduced the abundance of the pathogenic *Faecalibacterium* in the gut of piglets. Moreover, the metabolic functions affecting the digestive, endocrine, and excretory systems, which influence diarrhea in piglets, were enhanced significantly by the addition of SAEW. Correlation analysis showed that the influence of intestinal pathogens on the water and feed consumption and diarrhea of newly weaned piglets were relieved by the addition of SAEW. This work provides novel information with positive implications for the husbandry of diarrhea prevention in newly weaned piglets.

All the results of our study indicate that the antidiarrheal effect of SAEW on newly weaned piglets is closely related to decreased intestinal pathogens and enhanced bacterial metabolic functions. We recommend that SAEW could be used for the prevention of diarrhea in piglets during the first two weeks after weaning. The effects of SAEW on piglets who are suffering from diarrhea are, however, still unclear and should be the focus of future research.

## 6. Patents

A Chinese invention patent resulted from the work reported in this manuscript: A drinking water additive with health care effects for piglets and its use (ZL201810066437.3).

## Figures and Tables

**Figure 1 genes-14-01398-f001:**
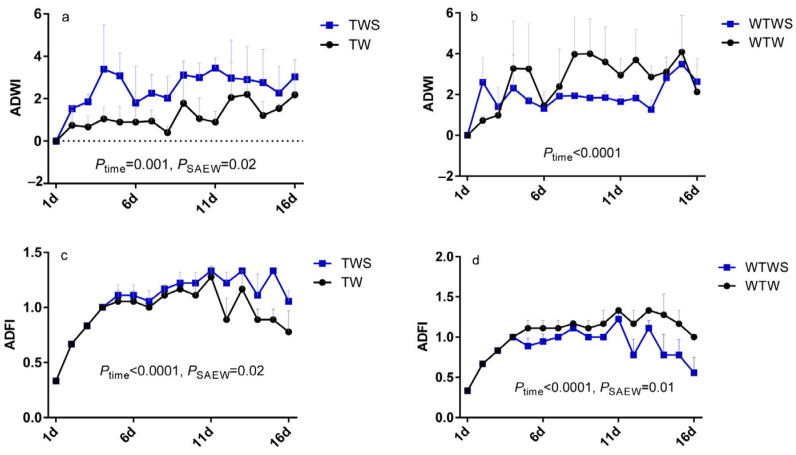
Water and feed consumption of piglets during the first 16 days: (**a**) ADWI comparison between TW and TWS; (**b**) ADWI comparison between WTW and WTWS; (**c**) ADFI comparison between TW and TWS; (**d**) ADFI comparison between WTW and WTWS. ADWI: average daily water intake; ADFI: average daily feed intake.

**Figure 2 genes-14-01398-f002:**
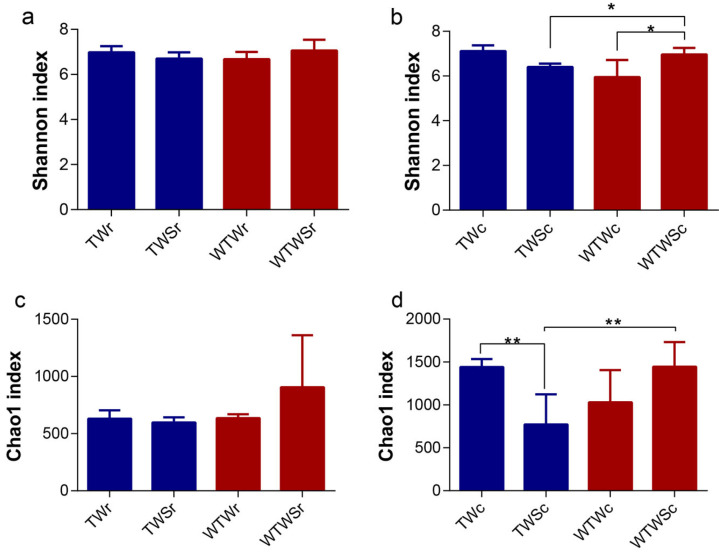
α diversity differences between groups: (**a**) Shannon index in rectum; (**b**) Shannon index in caecum; (**c**) Chao1 index in rectum; and (**d**) Chao1 index in caecum of piglets. * *p* < 0.05, ** *p* < 0.01, Mann-Whitney *U* test. TWr: samples in rectum with TW treatment; TWSr: samples in rectum with TWS treatment; WTWr: samples in rectum with WTW treatment; WTWSr: samples in rectum with WTWS treatment; TWc: samples in cecum with TW treatment; TWSc: samples in cecum with TWS treatment; WTWc: samples in cecum with WTW treatment; WTWSc: samples in cecum with WTWS treatment.

**Figure 3 genes-14-01398-f003:**
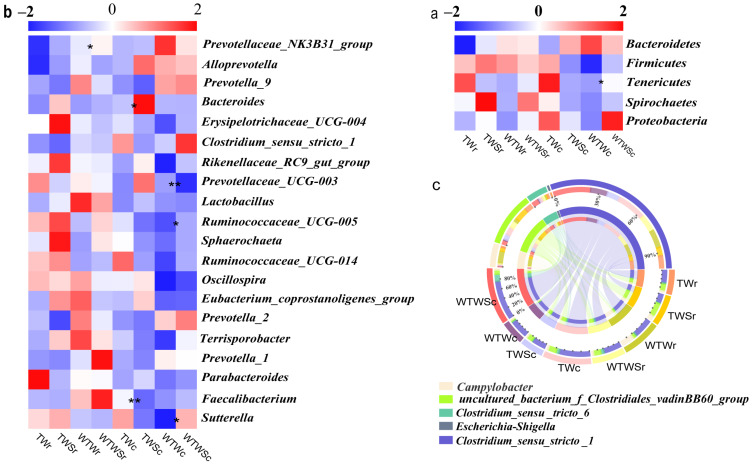
The intestinal pathogenic bacteria: (**a**) at the phylum level; (**b**) top 20 at the genus level; (**c**) five potential diarrhea associated genera. * *p* < 0.05, ** *p* < 0.01, Welch’s *t*-test. TWr: samples in rectum with TW treatment; TWSr: samples in rectum with TWS treatment; WTWr: samples in rectum with WTW treatment; WTWSr: samples in rectum with WTWS treatment; TWc: samples in cecum with TW treatment; TWSc: samples in cecum with TWS treatment; WTWc: samples in cecum with WTW treatment; WTWSc: samples in cecum with WTWS treatment.

**Figure 4 genes-14-01398-f004:**
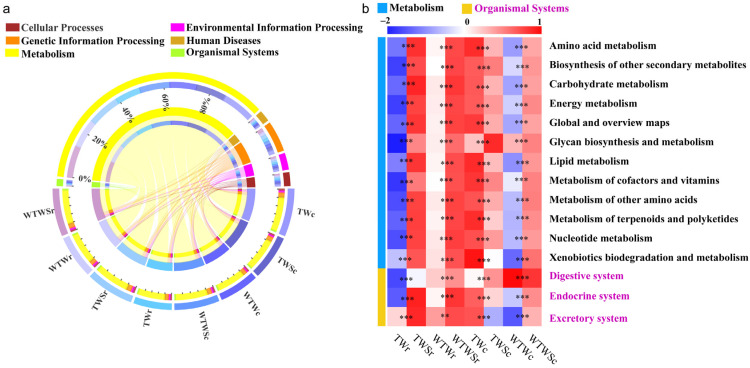
The metabolic pathways. (**a**) at the first level; (**b**) 15 functional metabolic pathways related to diarrhea of piglets. ** *p* < 0.01, *** *p* < 0.001, Welch’s *t*-test. TWr: samples in rectum with TW treatment; TWSr: samples in rectum with TWS treatment; WTWr: samples in rectum with WTW treatment; WTWSr: samples in rectum with WTWS treatment; TWc: samples in cecum with TW treatment; TWSc: samples in cecum with TWS treatment; WTWc: samples in cecum with WTW treatment; WTWSc: samples in cecum with WTWS treatment.

**Figure 5 genes-14-01398-f005:**
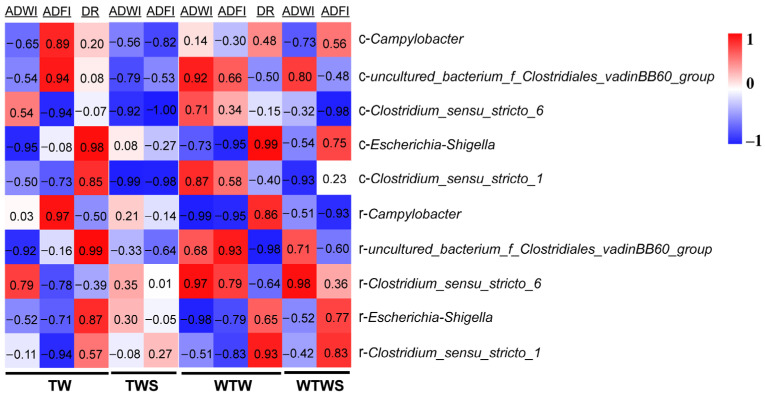
The correlation between the five potential diarrhea-associated genera and water and feed consumption of piglets. ADWI: average daily water intake; ADFI: average daily feed intake; DR: diarrhea rate. “c-” before the bacteria name means bacteria in caecum; “r-” before the bacteria name means bacteria in rectum.

**Table 1 genes-14-01398-t001:** The parameters of drinking water.

Groups	Total Bacteria (CFU/mL)	Coliforms (CFU/100 mL)	pH Value
TW (tap water)	215 ± 2.50 ^b^	110 ± 0 ^b^	6.89 ± 0.01
TWS (TW + 3% SAEW)	25 ± 1.32 ^d^	0 ^c^	6.80 ± 0
WTW (warm tap water, 30 °C)	>300 ^a^	120 ± 10 ^a^	6.85 ± 0.02
WTWS (WTW + 3% SAEW)	90 ± 3.12 ^c^	0 ^c^	6.85 ± 0.01

Values within the same column followed by different superscript lowercase are significantly different (*p* < 0.05).

**Table 2 genes-14-01398-t002:** Effects of different treatments on growth performance and health status of weaned piglets.

Groups	ADG (kg/d/Pig)	FCR	DR (%)
TW (tap water)	0.44 ± 0.04	1.74 ± 0.09	1.35 ± 0.58 ^a^
TWS (TW + 3% SAEW)	0.49 ± 0.05	1.68 ± 0.12	0 ^c^
WTW (warm tap water, 30 °C)	0.45 ± 0.05	1.73 ± 0.10	0.51 ± 0 ^b^
WTWS (WTW + 3% SAEW)	0.43 ± 0.02	1.68 ± 0.03	0 ^c^

ADG: average daily gain; FCR: feed consumption rate; DR: diarrhea rate. Values within the same column followed by different superscript lowercase are significantly different (*p* < 0.05).

## Data Availability

The 16S rRNA gene sequences determined in this study were deposited in the NCBI Sequence Read Archive (SRA) database with the accession number SRP149034.

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
