# Peer review of "Effect of Slightly Acidic Electrolyzed Water on Growth, Diarrhea and Intestinal Bacteria of Newly Weaned Piglets"

_genes, 2023, doi:10.3390/genes14071398_

Round 1
Reviewer 1 Report
The authors of the article explored the interesting topic of preventing diarrhea in piglets by adding slightly acidic electrolyzed water (SAEW) to the drinking water. 72 piglets were used for the study. Materials and methods are clearly and thoroughly described. 3% SAEW was added to the drinking water. The number of bacteria and coliforms in drinking water was reduced by 70% and 100%, respectively. The average daily water and feed intake of the piglets increased in the first 16 days, and the rate of diarrhea decreased by 100%, with not a single case of diarrhea recorded at the end of the experiment. These are very important observations that can avoid the administration of antibiotics, support antibiotic therapy, and thus avoid piglet deaths or at least reduce the cost of treatment. The authors further found that SAEW reduced the diversity of cecal bacteria when supplied with normal tap water and increased the richness of beneficial cecal bacteria when supplied with warm tap water.
Charts and tables are legible, statistical analysis shows statistically significant differences. The literature review contains current work on this topic.

Author Response
Thanks a lot for the reviewer's comments.
Reviewer 2 Report
Authors investigated the effects of slightly acidic electrolyzed water (SAEW) in combination with the tap or the warm tap water on the diarrhea prevention and the microbiome composition in piglets. The manuscript is well written, with the proper experimental design and statistics applied. I recommend publications without changes.
Author Response

(The authors gave the same response as above.)

Reviewer 3 Report
Effect of Slightly Acidic Electrolyzed Water on Growth, Diarrhea and Intestinal Bacteria of Newly Weaned Piglets
The present work aims to find a valid and environment-friendly method to improve the microbiological characteristics of piglet drinking water. Four groups of piglets were made up, involving a total of 72 piglets. Each group was given tap water, warm tap water, tap water with 3% HCl and warm tap water with 3% HCl. Water analysis, clinical observation with regard to diarrheal events and both colon and fecal matter sampled from the sacrificed piglets confirm the hypothesis. An improved microbiological quality of the water was assessed, in particular reducing greatly the number of coliforms (to 0), the number of diarrhea cases (idem) and improving the composition of the microbial communities of individuals which received water supplemented with SAEW. This study represents a very interesting option to improve swine management after colistin and Zn banning.
However, here is listed some suggestions for the Authors:
Introduction: you should add a sentence reassuming the microbiological characteristics of drinking water for pig herds according to Chinese law (or, if you prefer, in general)
Line 137: Chyme is the partially digested feed that pass from the stomach to the duodenum, but it changes its name throughout its path along the intestine. In particular, when in rectum, it is already called “feces”. Please check for the use of this term.
Line 137: digesta and feces sampling is not clear, what do you mean when saying that “A total of 24 chyme samples (12 rectums, 12 cecum, and three samples from each group) were collected and processed for sequencing.”?
Line 138: kits extract all the DNA found in the sample (genomic, bacterial, viral, and parasite DNA)
Line 162: Mann-Whitney test is normally used for statistical analysis, thus for comparing indices values. How did you exactly used this test?
Line 325: Shigella
Lines 329-331: please review this sentence, is not clear
First Heatmap: Tap water in samples from cecum seems to have enhanced metabolic pathways as well as modified water had. Could you discuss these data?
Author Response
Introduction: you should add a sentence reassuming the microbiological characteristics of drinking water for pig herds according to Chinese law (or, if you prefer, in general)
The total E. coli populations less than 100MPN/100ml and 10MPN/100ml in adult and young livestock drinking water, respectively, has been provided according to " Drinking Water Quality for Livestock and Poultry - Pollution free Food (NY5027-2008)" in China.
The above sentence has been added in introduction.
Line 137: Chyme is the partially digested feed that pass from the stomach to the duodenum, but it changes its name throughout its path along the intestine. In particular, when in rectum, it is already called “feces”. Please check for the use of this term.
The term “feces” has been added in Line 231, 297, 320 for its use.
Line 137: digesta and feces sampling is not clear, what do you mean when saying that “A total of 24 chyme samples (12 rectums, 12 cecum, and three samples from each group) were collected and processed for sequencing.”?
It means 12 samples were from rectum, and another 12 samples were from cecum. Four treatments were setup. So, 3 samples were from rectum, and another 3 samples were from cecum in each treatment (group).
Line 138: kits extract all the DNA found in the sample (genomic, bacterial, viral, and parasite DNA)
Yes. Then only bacterial genes were detected by 16S rRNA sequencing.
The sentence has been revised.
Line 162: Mann-Whitney test is normally used for statistical analysis, thus for comparing indices values. How did you exactly used this test?
This test was used for comparing the Shannon and Chao1 index differences between groups such as, TW and TWS.
The sentence has been revised in Line 164-166.
Line 325: Shigella
The word has been corrected throughout the manuscript.
Lines 329-331: please review this sentence, is not clear
The sentence has been revised as following:
The drinking water disinfection test indicated that warm water (30oC) carried more bacteria, and 3% SAEW with an available chlorine concentration (ACC) of 30 mg/l can be used as an efficient anti-bacteria agent for drinking-water of piglets.
First Heatmap: Tap water in samples from cecum seems to have enhanced metabolic pathways as well as modified water had. Could you discuss these data?
We are sorry for cannot find the enhanced metabolic pathways in tap water samples from cecum. All the data were comparing between tap water and modified water.